# Early Palliative Care in Acute Myeloid Leukemia

**DOI:** 10.3390/cancers14030478

**Published:** 2022-01-18

**Authors:** Leonardo Potenza, Eleonora Borelli, Sarah Bigi, Davide Giusti, Giuseppe Longo, Oreofe Odejide, Carlo Adolfo Porro, Camilla Zimmermann, Fabio Efficace, Eduardo Bruera, Mario Luppi, Elena Bandieri

**Affiliations:** 1Hematology Unit and Chair, Azienda Ospedaliera Universitaria di Modena and Department of Medical and Surgical Sciences, University of Modena and Reggio Emilia, 41125 Modena, Italy; eleonora.borrelli@unimore.it (E.B.); davide.giusti@unimore.it (D.G.); mario.luppi@unimore.it (M.L.); 2Department of Linguistic Sciences and Foreign Literatures, Catholic University of the Sacred Heart, 20123 Milan, Italy; sarah.bigi@unicatt.it; 3Oncological Medicine Unit, Azienda Ospedaliera Universitaria di Modena, 41125 Modena, Italy; longo@unimore.it; 4Department of Medical Oncology, Dana-Farber Cancer Institute, Boston, MA 02215, USA; Oreofe_Odejide@dfci.harvard.edu; 5Department of Biomedical, Metabolic and Neural Sciences, University of Modena and Reggio Emilia, 41125 Modena, Italy; carlo.porro@unimore.it; 6Center for Neuroscience and Neurotechnology, University of Modena and Reggio Emilia, 41125 Modena, Italy; 7Princess Margaret Cancer Centre, University Health Network, Toronto, ON M5B 2M9, Canada; Camilla.Zimmermann@uhn.ca; 8Division of Palliative Care, University of Toronto, Toronto, ON M5S 1A4, Canada; 9Italian Group for Adult Hematologic Diseases (GIMEMA), Health Outcomes Research Unit, 00182 Rome, Italy; f.efficace@gimema.it; 10Palliative Care & Rehabilitation Medicine, University of Texas MD Anderson Cancer Center, Houston, TX 77030, USA; ebruera@mdanderson.org; 11Oncology and Palliative Care Units, Civil Hospital Carpi, Local Health Agency, 41012 Carpi, Italy; e.bandieri@ausl.mo.it

**Keywords:** acute myeloid leukemia, early palliative care, integration, disease trajectory

## Abstract

**Simple Summary:**

Several new targeted drugs for the treatment of acute myeloid leukemia (AML) have been developed in recent years. However, their potential efficacy should be balanced with the possibility of a poor outcome. For patients with solid tumors, the integration of palliative care (PC) with standard oncologic care, eight weeks after the diagnosis of advanced cancer, has demonstrated many benefits. In patients with hematologic malignancies, this model is underutilized. Here, we review the PC needs and the evidence in the literature supporting the early integration of PC in the disease trajectory of patients with AML. Early PC should be the standard of care for patients with AML. Efforts should also be made to extend the integrated PC model to patients with other hematologic malignancies.

**Abstract:**

Background: Several novel targeted therapies seem to improve the outcome of acute myeloid leukemia (AML) patients. Nonetheless, the 5-year survival rate remains below 40%, and the trajectory of the disease remains physically and emotionally challenging, with little time to make relevant decisions. For patients with advanced solid tumors, the integration of early palliative care (EPC) with standard oncologic care a few weeks after diagnosis has demonstrated several benefits. However, this model is underutilized in patients with hematologic malignancies. Methods: In this article, we analyze the palliative care (PC) needs of AML patients, examine the operational aspects of an integrated model, and review the evidence in favor of EPC integration in the AML course. Results: AML patients have a high burden of physical and psychological symptoms and high use of avoidant coping strategies. Emerging studies, including a phase III randomized controlled trial, have reported that EPC is feasible for inpatients and outpatients, improves quality of life (QoL), promotes adaptive coping, reduces psychological symptoms, and enhances the quality of end-of-life care. Conclusions: EPC should become the new standard of care for AML patients. However, this raises issues about the urgent development of adequate programs of education to increase timely access to PC.

## 1. Introduction

For more than 40 years, the treatment of acute myeloid leukemia (AML) in patients younger than age 60 has been mainly based on classical conventional chemotherapy schedules, the so-called 3 + 7 schemes, including cytarabine and an anthracycline, or similar protocols [1,2]. Post-remission therapies have included consolidation and maintenance chemotherapies for patients with favorable disease-related features and hematopoietic stem cell transplant (HSCT) in first remission for patients with adverse-risk disease if they can tolerate the procedure. In patients older than 65 years, non-intensive strategies with hypomethylating agent monotherapy have been preferred because of a favorable balance between treatment-related mortality or morbidity and efficacy [1,2]. The former approach is still associated with a high frequency of relapse, with long-term survival achieved in only 35–45% of patients. The latter does not affect the disease course, as more than 70% of older patients die within 1 year of diagnosis, and long-term survival occurs in less than 10% of them [1,2].

Recent years have witnessed a new wave of drug approvals, which has increased the treatment options for patients with refractory/relapsed disease and older, unfit populations. The number of targeted therapies recently licensed or tested for specific mutations in AML has resulted in encouraging improvements in clinical outcomes and offers a unique treatment opportunity for specific AML patient populations [3,4]. Although these emerging therapies seem to improve remission and survival, outcomes for most patients remain dismal, and the disease trajectory is highly physically and emotionally challenging and filled with a high level of uncertainty.

In addition, this wave of new therapies is accompanied by increasing complexity in care delivery. For example, patients may be exposed to a unique array of new adverse event profiles, which may hamper the quality of life (QoL) in the face of limited survival advantages [3,4]. The occurrence of tumor lysis syndrome and infections related to profound and prolonged cytopenia with venetoclax, the development of differentiation syndrome with gilteritinib and isocitrate dehydrogenase (IDH) inhibitors, and QTc prolongation with IDH inhibitors can all cause severe or life-threatening symptoms [1,2,3,4]. Patients are thus faced with the difficult task of balancing the pursuit of prolonged survival with the risk of increased hospitalizations and suffering. Similarly, hematologists have to appropriately weigh whether the expected benefits of novel targeted agents and their safety profiles are aligned with patients’ values, wishes, and desired QoL to help the patients themselves share in the decision making.

Such complexities of balancing the efficacy of new and promising therapies with the possibility of a poor outcome raise potential concerns regarding the quality of care that AML patients receive, including physical and emotional care, shared decision making, and end-of-life care.

In patients with solid tumors, an innovative integrated model of assistance, in which palliative support is combined with standard cancer care as soon as the patient is diagnosed with incurable and/or advanced disease, has been extensively studied. This model has shown many benefits, including improved management of disabling physical and psychological symptoms; facilitation of coping and assistance in sharing goals among patients, families, and medical staff; and improved QoL [5]. However, this model is underutilized in patients with hematologic malignancies [6].

In this review, we describe the disease experience and palliative care needs of patients with AML. We also aim to evaluate whether the existing literature supports the early integration of palliative care for patients with AML and their caregivers in the current challenging clinical landscape.

## 2. AML Patients’ Physical Symptoms

Just after receiving the diagnosis, patients with AML must cope with difficult decisions.

Additionally, even before starting therapy, AML patients have a substantial burden of fatigue, which is clinically meaningfully worse than that of their peers from the general population [7]. The circumstances are even harder for patients with refractory/relapsed disease and for those older than 60 years. Fit patients may have to choose between in-hospital intensive chemotherapy and in-clinic less-intensive therapy, while unfit and older patients may choose between in-clinic less-intensive therapy and supportive therapy. Inpatient induction chemotherapy requires patients to leave their life behind, with only hours or days of notice, for a period of at least 3–4 weeks. The progression of the disease and the initiation of treatment also determine the occurrence/worsening of physical symptoms [8,9].

Patients with hematologic malignancies, including those with AML, appear to have a symptom severity equal to if not worse than solid tumor patients. Two cross-sectional studies that enrolled 180 and 108 patients with hematologic malignancies, of whom more than 30% and 10% had AML, respectively, reported levels of dyspnea, nausea, and anorexia similar to those suffered by patients with advanced solid cancers [8,9]. Patients with refractory disease had the highest mean number of symptoms and level of physical distress. Approximately 50–70% of patients reported a lack of energy, feeling worried, difficulty sleeping, drowsiness, dry mouth, and feeling sad as the most common symptoms. From 40% to 50% of them suffered from constipation, difficulty swallowing, problems with urination, loss of sexual interest, swollen arms and/or legs, and hair loss. Several symptoms (e.g., lack of energy, drowsiness, dry mouth, feeling nervous, lack of appetite, sweats, mouth sores, feeling worried, cough, and weight loss) occurred more frequently among inpatients than outpatients. Moreover, patients with hematologic malignancies had statistically higher rates of clinically significant tiredness and drowsiness compared with solid tumor patients (51% vs. 42%; *p* = 0.03; and 30% vs. 20%; *p* = 0.05). The odds of being tired and drowsy were at least twice as high as solid tumor patients, and they were correlated with age, gender, race, and performance status (OR 2.19; 95% CI 1.22–3.91; *p* < 0.01; OR 1.81; CI 1.07–3.07; *p* < 0.03) [8,9]. Another study, using the Memorial Symptom Assessment Scale (MSAS), found that among 249 patients with acute leukemia, of whom 193 (78%) had AML and were one month from either diagnosis or relapse, the median number of physical and psychological symptoms was 9 and 2, respectively. Additionally, 91% and 61% of the patients reported >5 concurrent physical and psychological symptoms and >10 concurrent symptoms, respectively. The most intense symptoms were severe lack of energy, difficulty sleeping, and pain [10].

Indeed, patients with hematological malignancies, particularly those with AML, may also experience pain, which may afflict as many as half of patients, with severity ranging from slight in 8%, moderate in 25%, to severe in up to 35% [10,11,12]. Two more studies, the first using the Edmonton Symptoms Assessment System (ESAS) and the second using the MSAS, among 40 and 14 AML patients, respectively, demonstrated that pain in the early phase of diagnosis was higher than in patients with solid tumors in the same phase. In addition, the pain that patients with AML experienced was reported to increase over the first six weeks after diagnosis, remaining one of the most severe and distressing symptoms during the entire induction chemotherapy course [13,14].

## 3. AML Patients’ Illness Experience

The staggering occurrence of AML, the associated physical limitations, and a clinical course characterized by a threat to life that arises either immediately or at several time-points during the disease trajectory increase the possibility that patients will experience significant emotional and psychological distress. Several studies have demonstrated that patients with acute leukemia have one of the highest levels of distress among cancer patients and more frequently show clinically significant symptoms of post-traumatic stress disorder [15,16].

In four qualitative studies that enrolled 129 AML patients overall, the diagnosis of AML was consistently described as traumatic and shocking [17,18,19,20]. It was so overwhelming that it negatively impacted either the understanding of available therapies or the acquisition of all information about the disease and impaired making treatment decisions [17,18,19,20]. The patients defined the nature of leukemia and the outcome of treatments as being unpredictable and reported maintaining high levels of uncertainty regarding the prognoses and the next steps in care throughout the entire course of the disease [17,19,20].

Quantitative data showed that, despite the documentation in the medical record, more than 75% of patients denied being offered treatment options other than the one that they chose [18]. Moreover, two weeks after the diagnosis, they reported psychological symptoms with frequently or at least occasionally diminished quality of life (QOL) scores, especially when undergoing intensive chemotherapy, and generally poor scores of emotional well-being (EWB) [10].

Most of those feelings and sentiments have been reported to persist even three months after the diagnosis [21]. In one study, AML patients reported psychological difficulties, including helplessness and hopelessness, a high level of stress, and isolation, which compounded their existing relevant physical symptoms, such as worsening low energy levels, feelings of weakness or tiredness, and restrictions in daily activities [20]. Their personal interests or hobbies and the devotion of their family members or friends represented the only relief, although the burden that the caregivers bore was a great concern [17,19,20]. Moreover, AML patients described reassurance by the trust that they developed in the medical team to which they offered control of the situation. Patients reported that all of the unexpectedness induced by the diagnosis led them to defer to the hematologists to make decisions about treatment. They preferred receiving information gradually to avoid feeling overwhelmed and to maintain focus on the immediate initial treatment, avoiding thoughts about subsequent admissions and/or prognosis [17,18,19,20]. Indeed, in two further studies, 52 AML patients expressed the need for practical information mainly focused on dealing with chemotherapy side effects and how treatment would limit daily living, and almost 90% of them felt that they were given an adequate amount of information about those topics [19,21].

In patients with solid tumors, it has been reported that such a coping strategy partially reduced anxiety, yet it increased the sense of uncertainty about what to expect from treatment, what the trajectory of the disease is, and what the future will bring, leading to eventual frustration [22]. Consistent with this, it has been reported that partial information about the trajectory of the disease may result in an overestimation of patients’ chances of cure and one-year survival. In two studies, about one-third to half of 31 AML patients aged 50 or older thought that there was a 75–90% chance of survival, regardless of the therapy that they chose, and among 100 AML patients aged ≥60 years, 98% of those receiving intensive chemotherapy and 82% of those undergoing non-intensive chemotherapy reported that they were “somewhat” or “very likely” to be cured of their disease, while their hematologist estimated that this applied to only 31% of patients [23,24]. Finally, in a recent work, more than half of 50 AML patients with a mean age of 63 years reported that treatment had a high chance of cure, 22% felt that the treatment goal was to live longer, and only 2% recognized that the treatment was aimed at the palliation of symptoms. Overall, only one-third agreed with their estimated prognosis. The agreement was more frequently found in patients with favorable-risk disease than patients with unfavorable-risk AML [25].

Remarkably, patients with AML who better understood their prognosis by recognizing a lower likelihood of cure reported significantly higher depression symptoms, as measured by the Hospital Anxiety and Depression Scale (HADS), a trend towards lower QOL based on the Functional Assessment of Cancer Therapy–Leukemia questionnaire (FACT-Leuk) at 1 month after study enrollment, and significantly worse EBW scores, with a median EWB score of 12 compared to a score of 19 for AML patients who did not agree with prognosis estimates and other cancer patients [24,25]. Of note, in lung and gastrointestinal cancer patients, it has been reported that the use of certain coping strategies may buffer these relationships in patients who have an accurate estimate of their prognosis [22]. All of these data suggest that interventions to develop more adaptive coping strategies should be strongly encouraged in AML patients to improve their prognostic awareness as well as their ability to cope with this prognosis.

## 4. Healthcare Utilization and End-of-Life Care in Patients with AML

Many patients with hematologic malignancies have an unpredictable illness course, with the possibility of cure persisting even in the relapsed and refractory setting, in contrast to patients with advanced solid tumors. Five-year survival for AML patients after first relapse is about 10–30%, and it may increase to approximately 40–50% when an HSCT is performed after a myeloablative conditioning regimen [26]. Moreover, the changes in a patient’s disease status and the decline near death can be rapid with hematologic malignancies, as the disease trajectory is unpredictable and lacks a clear transition between the curative and palliative phases of treatment. Hematologic oncologists report that this impacts their ability to accurately determine when a patient is at the end of life (EOL) and reduces the possibility of discussing prognostic information with their patients [27].

The challenges of the disease course and physicians’ evaluations, along with patients’ frequent overestimation of their prognosis and goals of treatment, have relevant clinical implications, including negatively impacting patients’ ability to make informed decisions and their likelihood of receiving appropriate EOL care. Indeed, overestimation of the likelihood of cure and survival is associated with patients’ higher willingness to either accept further disease-oriented treatment or opt for intensive medical care at the EOL and with hematologists’ higher likelihood of prescribing systemic therapy with no survival benefit for patients with either Eastern Cooperative Oncology Group performance status 3 and 4 or an expected survival of 1 month [28,29].

Significant differences in the quality of EOL indicators have been reported between patients with hematologic malignancies and those with solid tumors [30]. In the last month of life, patients with hematologic malignancies were more likely than their solid tumor counterparts to receive either chemotherapy or targeted therapy, to visit the emergency room, to be admitted to the hospital or intensive care unit (ICU), to have several days of hospitalization, and to die either in the hospital or ICU [27,28,30]. This was in contrast to the fact that a large sample of hematologists considered the majority of measures included in the list of standard EOL quality measures to be “acceptable” [31,32].

Indicators of intense healthcare utilization near the EOL are especially frequent for patients with AML. Data from the Surveillance, Epidemiology and End Results (SEER) Medicare linked database and from single-center retrospective or prospective studies have reported that, from 1999 to 2012, more than 10% and almost 50% of AML patients still received chemotherapy in the last 14 and 30 days of life, respectively. Of these patients, 30–50% were admitted to the ICU, and 88.9–92.3% were admitted to the hospital in the last 30 days of life [33,34,35,36]. Of note, two studies measuring the time spent in the hospital demonstrated that AML patients devoted more than 40% of their life after diagnosis either in the hospital or attending outpatient clinic appointments and spent an average of 21.4 days in their last month of life hospitalized [34,35,36]. These figures are even more worrisome if we consider the results of a retrospective study reporting that 172 (27%) of 649 unplanned hospital admissions were potentially avoidable in AML patients from two tertiary care hospitals in the USA [37]. Strikingly, almost 50% of these admissions were due to problems that were potentially manageable in the outpatient setting or the failure of timely outpatient follow-up, or they occurred for patients receiving either hospice or supportive care [37].

Several studies, including another SEER-Medicare analysis, have demonstrated that although the rates of hospice enrollment for AML patients have increased from 30% in 1999 to 56.4% in 2012, the median duration of hospice stay was still 9 days and was even shorter in transfusion-dependent patients, namely, 6 days [37]. Moreover, the first hospice enrollment started in the last 7 and 3 days of life for 47.3% and 28.8% of patients, respectively [38]. Overall, 43–69.8% of AML patients have been reported to still die in hospitals [33,34,35,36,37,38,39].

Of note, although goals of care (GOC) discussions can improve the quality of EOL care, many patients with hematologic malignancies still receive intensive care at the EOL despite almost two-thirds having documented GOC discussions. These data suggest that GOC discussions alone are not sufficient, per se, to improve the quality of EOL care and that other factors—such as the timing, location, and quality of these discussions—should be evaluated [40]. Moreover, the availability of palliative care (PC) programs appears to only modestly influence the above-mentioned figures, because less than one-fifth to no more than one-third of patients have been reported to receive a consultation, and because the median time from the consultation to death was no longer than 7 days [34,35,36,39,41]. These latter data strongly suggest that more effective methods of providing palliative care for patients with hematologic malignancies should be identified.

## 5. Early Palliative Care: Lessons from Advanced Solid Cancer Patients

Several randomized controlled trials (RCTs) have provided level I evidence about the efficacy of the integration of palliative care from the time of diagnosis in patients with advanced solid tumors [42,43,44,45,46]. This integrated model was associated with statistically significant improvement in patients’ QOL in almost all of the studies; improvement in mood in five of them; reduction in symptom burden in three studies; increased satisfaction with care, earlier and longer hospice referral, and reduction in aggressiveness at the EOL in one study each; and, in two of the studies, longer survival [42,43,44,45,46,47]. Additionally, three trials also reported caregivers’ increased satisfaction with care, improvement in psychological symptoms, and lower depression and stress burden, but no significant differences in their QoL [48,49,50]. Of note, no adverse outcomes were reported for early palliative care (EPC) involvement in any of these studies. The results of these trials have been supported by many prospective cohorts and retrospective studies, which have replicated the many benefits associated with the integration of EPC into the oncological care of patients with advanced-stage cancer [51,52].

Finally, a Cochrane review and a recent meta-analysis confirmed that EPC integration into standard oncologic care may have more beneficial effects on QoL, symptom intensity, and overall survival when compared to standard oncologic care alone in patients with advanced solid malignancies [53,54].

Based on the results of all of these studies, the American Society of Clinical Oncology has recommended that patients with metastatic solid cancer be offered concurrent palliative care with standard oncologic care early during the trajectory of the disease, possibly as early as the initial diagnosis [55].

## 6. What Does the Intervention Consist of?

The clinical members forming the team, the frequency of visits, the location of care, and the duration of the EPC intervention may vary among studies. However, all of the described interventions share the “reflective, intimate and practical” characteristics of developing strong relationships with patients and caregivers and management of the profound physical and emotional concerns related to the disease status, and they may, ultimately, cover end-of-life issues when required by the illness trajectory and clinical situation [56]. Indeed, the intervention can be delivered by board-certified palliative care physicians and advanced-practice nurses, as well as single specialized palliative care nurses or a full palliative care team composed of a specialized physician, nurse practitioner, social worker, and chaplain [42,43,44,45,46,47,48,49]. EPC may consist of initial inpatient evaluation and then monthly outpatient visits and, as needed, on request, outpatient evaluations with or without the availability of a 24 h on-call service for the telephone management of urgent issues, weekly telephone-based nursing-led multicomponent psycho-educational sessions, or even a comprehensive palliative care consultation by the full inpatient team on the same or following day of the emergency department admission, with subsequent follow-up visits [42,43,44,45,46]. Overall, patients assigned to EPC receive assessment and treatment of physical and psychosocial symptoms, assistance with decision making regarding treatment, evaluation of coping with their illness, coordination of care based on their individual needs, support for caregivers, establishment of goals of care and advance care plans, and plans for the transition phase by assessing patients’ understanding of disease status, treatment efficacy, illness trajectory, and prognosis. Of note, the cultivation of prognostic awareness of patients is considered one major component of the EPC intervention [42,43,44,45,46,47,48,49,56,57]. A recent study summarized and delineated the complex EPC intervention delivered in the clinic. It was defined as “team-based outpatient early palliative care” to assist specialized physicians, researchers, and oncologists in implementing, replicating, and adapting it for various needs of either patients or institutions. Four domains and four principles constitute the conceptual framework. The domains include coping and support, symptom control, decision making, and future planning. The key principles refer to flexible, attentive, patient-led, and family-centered care [58].

## 7. Early Palliative Care in Patients with Hematologic Malignancies

Numerous studies have identified that patients with hematologic malignancies, especially those with AML, and their caregivers, are burdened by physical and psychological symptoms and have many unmet palliative care needs. However, most reports available in the literature have focused mainly on the feasibility and acceptability of integrating palliative care into the treatment plan of patients undergoing HSCT or those affected by multiple myeloma (MM). In the first study, a PC service collaborated with the HSCT unit in a tertiary care center. The PC team provided 392 consultations to 256 patients. The main reason for the consultation was pain, which was reported in 278 (71%) out of 392 PC visits. The PC physicians also provided the first goals of care conversations in 172 (67%) patients. The collaboration also reported that the hospice referral rate of dying patients increased from 5% pre-implementation to 41% after the implementation. The hematologists reported high levels of satisfaction with the program [59].

A second study evaluated consultations conducted by PC-trained nurses, which occurred before hospital admission for either autologous or allogeneic HSCT and then monthly during the entire procedure. Patients’ self-reported assessments were collected at baseline and 60 and 90 days post-transplantation. Of 32 patients, 20 (62.5%)—75% affected by acute leukemias—consented to participate. Of the 17 patients (85%) who completed the EPC survey, 14 (82%) reported being very comfortable, and 3 (18%) were comfortable with the EPC evaluation. Although there was no statistically significant change in mood (assessed by the HADS), four (24%) reported a reduced level of anxiety, and two (12%) reported feeling more hopeful after the EPC intervention. The authors concluded that pre-transplant EPC consults were feasible and acceptable and did not negatively affect mood or hope. Of note, patients stated that of the 14 caregivers present at the initial consultation, 8 were very comfortable, and 6 were comfortable [60].

A third study evaluated the offer of a palliative care program, delivered during office hours, for hematologic oncology services. PC nurse practitioners provided the service by discussing a mean of 11 patients per week with hematologists. Pain was the topic most discussed, followed by nausea, anxiety, and depression. Of these patients, 14.7% subsequently needed a full PC consult. Consults for GOC discussions significantly increased. The program was judged feasible and well accepted [61].

A fourth study retrospectively investigated the effect of PC consultations in MM patients. In less than one year, 67 (40%) of all 169 MM patients admitted to the authors’ institution were referred to the PC service because they experienced disabling symptoms. The first PC consultation occurred almost one year after the diagnosis and was followed by three follow-up visits at two, three, and eight weeks after the initial consultation. Pain and mood disturbance were the main reasons for PC evaluation. The intervention was associated with statistically and clinically significant reduction in either the average or worst pain or the median physical and emotional symptom burden [62].

Lastly, the first RCT focused solely on patients with hematologic malignancies and demonstrated the benefit of EPC in HSCT patients. One hundred and sixty patients were randomized, seventy-nine receiving standard transplant care and eighty-one receiving inpatient PC intervention concurrent with standard transplant procedures [63]. A PC-trained physician or nurse practitioner provided the intervention by visiting the patient at least twice a week during hospitalization. The patients in the integrated PC modality had both a statistically and clinically significant lower decrease in QoL from baseline to two weeks, lower mean depression scores at two weeks and at three months, a decrease in anxiety syndrome at two weeks but not at three months, and lower increases in symptom burden from baseline to two weeks but not at three months, as measured by the FACT-BMT and HADS scales and the Patient Health Questionnaire 9 (PHQ-9). A subsequent analysis in the same study demonstrated sustained improvements in depression and post-traumatic stress disorder symptoms at six months post-HSCT in the intervention group [64]. Of note, the original study also enrolled 94 caregivers. Caregivers of EPC patients reported a lower increase in depression symptoms and improvements in coping and the administrative and financial domains of QOL at week 2 compared to those of controls, but no significant differences in overall QoL, anxiety, and depression [63]. This study had two further merits. Firstly, it demonstrated the advantages of a relatively low-dose intervention, represented by a median of eight visits per patient without follow-up after hospitalization. Secondly, the curative-intent setting of the study strongly suggests the prognosis-independent benefit of palliative care interventions.

Taken together, the results of all of the aforementioned studies support testing the efficacy of EPC in the population of hematologic patients most burdened by disease and treatments: those with AML.

## 8. Early Palliative Care in AML Patients

To date, only one randomized phase III, one randomized phase II, and one retrospective study have reported that the early integration of PC with standard hematologic care may be potentially effective for patients during the burdensome trajectory of AML (Table 1) [65,66,67].

El-Jawahri and colleagues were the first to demonstrate the benefits of early, integrated PC by conducting a multisite RCT. They enrolled 160 patients 18 years and older with high-risk AML receiving intensive chemotherapy. Patients were newly diagnosed, 60 years and older, either with a therapy-related disorder or with relapsed or primary refractory AML. Intensive chemotherapy was considered the 3 + 7 scheme, with cytarabine and an anthracycline and the possibility of additional drugs or other chemotherapy regimens requiring a 3- to 6-week hospitalization. In the intervention arm, an inpatient palliative care physician, an advanced practice nurse, or physician assistant saw the patient within 72 h of randomization, with at least two visits a week throughout the patient’s hospitalization for intensive chemotherapy and subsequent hospitalizations up to 1 year after randomization. PC clinicians did not see patients in the outpatient setting. Of note, the intervention was consistent with prior works of the group either in patients with solid tumors or in those with hematologic malignancies undergoing stem cell transplantation. The primary outcome of the study was QOL (based on the FACT-Leukemia scale) at week 2. Secondary outcomes included symptom burden employing the Edmonton Symptom Assessment Scale, depression and anxiety symptoms using PHQ-9 and the relative subscales of HADS, and PTSD symptoms using the PTSD Checklist–Civilian at week 2. Patients in the intervention arm received a median of 2.2 visits (range 2–5) per week. Patient-reported EOL discussions, hospitalizations in the last week of life, chemotherapy administration in the last 30 days of life, and hospice use were also considered to be EOL outcomes. Patients assigned to the PC arm reported statistically significantly better QOL, lower depression and anxiety, and reduced PTSD symptoms at week 2, and most of these results were maintained longitudinally [65]. Of note, a more recent analysis of the same patients revealed that patients randomized to EPC reported significantly greater use of approach-oriented coping and less use of avoidant coping at week 2 compared to those randomized to usual care, and the results indicated that these changes in coping mediated the intervention effects on QOL, depression, and anxiety symptoms. The indirect effect of coping accounted for 78%, 66.3%, and 35% of the total effect of EPC on QOL, depression, and anxiety, respectively [68].

Furthermore, another secondary analysis of the same patient cohort showed that 28% of AML patients suffered PTSD symptoms one month after the diagnosis. The symptoms were more likely associated with a lower baseline and a higher decline in QOL during the first 2 weeks of hospitalization in the multivariable regression analysis. Once more, the use of approach-oriented coping was associated with fewer PTSD symptoms at 1 month. The study is relevant because it emphasizes the critical need to screen patients with AML for PTSD symptoms by demonstrating, for the first time, that PTSD rates in this setting are significantly higher than those reported in other hematologic and cancer patients [69,70].

Symptom burden demonstrated similar results between the two groups. With respect to EOL outcomes, patients receiving EPC were more likely to report discussing EOL preferences and were less likely to receive chemotherapy in the last 30 days of life. There was no difference in hospice use, hospice length of stay, or hospitalization in the last week of life [65].

In a phase II trial, the authors developed a novel manualized intervention called Emotion And Symptom-focused Engagement (EASE) for patients with newly diagnosed or relapsed acute leukemia (AL), with the aim to evaluate its feasibility and tolerability and to determine its efficacy on traumatic stress symptoms and physical symptom burden. The study enrolled 42 AL patients, including 31 with AML. The intervention group received a psychotherapeutic intervention (EASE-psy), including 8–12 psychotherapeutic sessions, 30–60 min each, delivered over 8 weeks by a trained mental health clinician, and systematic screening of physical symptoms with targeted referral to palliative care (EASE-phys) with the ESAS-AL. The 17 AML patients receiving EASE demonstrated significant reductions in traumatic stress symptoms, clinically relevant and threshold acute stress disorders, pain intensity, and pain interference, compared with the 14 patients undergoing usual care. The study also reported a trend favoring EASE compared with the usual care on most of the other secondary outcomes, including QoL, depressive symptoms, and patient satisfaction [66].

In an observational retrospective study, an Italian group demonstrated the effects of EPC on the quality of EOL care in a real-life outpatient setting. The study enrolled 215 AML patients, including all risk groups but acute promyelocytic leukemia. Of the enrolled patients, 131 received EPC (within 8 weeks of AML diagnosis) and 84 received late PC [67]. The aim of the study was to assess the presence of 5 indicators of quality for palliative care and 10 indicators of the quality of EOL care [31]. Notably, the EPC intervention in this study was consistent with prior works of the group in patients with solid malignancies [51,52]. Moreover, according to the data from the literature and previous experiences, the EPC intervention was considered full when AML patients received three or more visits in the EPC clinic, as time is a relevant issue in providing an adequate and complete intervention [51,52,58]. Patients received the integrated model at a median of 5 weeks after the disease diagnosis either on the same day of the first outpatient visit for patients who had received inpatient intensive chemotherapy or on the first day of treatment for those receiving non-intensive chemotherapy or supportive care in the clinic.

EPC patients demonstrated high rates of receiving quality palliative care. All of them received assessment and management of pain, more than 70% had GOC discussions, and almost 60% had advance care planning conversations. In addition, 55% received psychological support and 43.5% received home care services. Importantly, patients who received EPC experienced a clinical and statistically significant reduction in pain. All of the figures were lower in patients receiving late PC. The high rates of quality palliative care were matched by low rates of aggressive healthcare utilization among deceased patients, with less than 3%, 10%, and 30% of EPC patients receiving chemotherapy in the last 14, 30, and 90 days of life, respectively. None of these patients was admitted to the ICU, nor did they receive CPR or intubations within the last month of life. More than 60% of decedents received home care service; however, only 10% had a hospice length of stay longer than one week, and the risk of dying at the hospital was 44.0% [67].

These studies demonstrate the successful integration of PC in patients with AML (Figure 1). Combined with the results of EPC in HSCT, the existing literature suggests that the integrated model of EPC could be extended to patients with other hematologic malignancies.

## 9. Future Challenges

The acknowledgment of the successful models of EPC discussed above raises several challenges that may spur future research. First, identification of other hematologic malignancy patients who may benefit the most from early access to PC is needed. Patients with multiple myeloma may be the most likely candidates because they have the most severe symptoms and their disease is largely incurable. Patients with lymphoma also stand to benefit from PC tailored to their disease phase or planned therapy, particularly with the evolving therapeutic landscape (e.g., chimeric antigen T-cell therapy (CAR-T) cell therapy, bi-specific or conjugated antibodies, targeted therapies), which is associated with unique side-effect profiles and heightened prognostic uncertainty [62,71,72,73]. Efforts are required to improve the knowledge of the PC needs of patients with hematologic malignancies other than AML and to evaluate the effects of PC intervention in other categories of patients, as available studies remain insufficient.

Second, the content of the intervention should be made clear. Although some detailed descriptions have been attempted, at least in solid cancer patients [58], the variability of EPC interventions remains high, especially in the hematologic setting. Future studies should conduct a comprehensive and consistent analysis of the tasks shaping such a complex intervention and the ideal timing for each task.

Third, the increasing need for PC integration may increase the request for palliative care physicians; in some countries, there is a shortage of trained palliative care physicians, while, in others, the construction of teaching organizations for an adequate program is still in progress.

Training programs that introduce a palliative care module to the training curriculum for oncology or hematology specialties and provide dual board-certified medical hematologists/oncologists and palliative care specialist physicians are thus needed. In addition, primary palliative care training efforts are essential to equip hematologists and oncologists with skills to recognize the PC needs of their patients, deliver primary palliative care to all of their patients, and effectively identify those patients requiring further specialist palliative care [74].

Fourth, a recent study showed that the Multinational Association for Supportive Care in Cancer guidelines seldom refer to the need to integrate PC into standard oncology care, especially in hematological cancers [75]. This could be one area where some improvement should be recommended, as it would be an easy goal to achieve.

Fifth, palliative care is complex and time demanding. Hematology/oncology teams face the burden of increasingly complex cancer treatments and a growing number of patients. Palliative care teams face the challenge of very limited resources [76]. Ultimately, the responsibility to provide patients access to early and fully integrated palliative care resides with the hospital executives and healthcare leaders. Thus, the structures are in place, the integration takes place, and patients achieve the outcomes of improved care, especially close to the end of life.

## 10. Conclusions

Patients with AML have many unmet palliative care needs, which continue to rise in this era of new targeted therapies. The emerging literature, including a phase III RCT, shows that the integration of palliative care in the AML disease course as soon as the diagnosis is made is feasible either during inpatient hospitalization or at the time of outpatient clinic evaluation. The integrated model may improve QoL, promote adaptive coping, reduce depression, anxiety, and PTSD symptoms, and enhance the quality of medical assistance and of EOL care. EPC should be advocated as the new standard of care for patients with AML. However, such a perception makes it imperative to expand studies supporting the integrated model and calls for the urgent development of adequate programs of education to increase and standardize PC among institutions or, at least, in tertiary care centers.

## Figures and Tables

**Figure 1 cancers-14-00478-f001:**
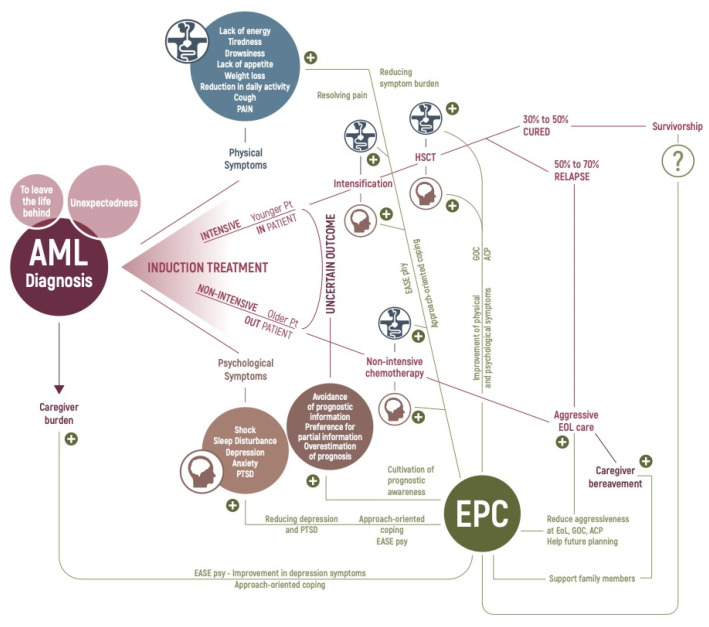
Effects of the early integration of palliative care on the disease trajectory of patients with acute myeloid leukemia. AML = acute myeloid leukemia; Pt = patients; PTSD = post-traumatic stress disorder symptoms; EASE = Emotion And Symptom-focused Engagement; psy = psychotherapeutic intervention; phy = screening of physical symptoms with targeted referral to palliative care; EPC = early palliative care; HSCT = hematopoietic stem cell transplant; GOC = goals of care discussions; ACP = advance care planning; EOL = end-of-life; ? = doubts about the possibility that EPC might be the best model for providing assistance and support to the population of AML patients cured of long-term therapy-related symptoms and mental health effects.

**Table 1 cancers-14-00478-t001:** Studies evaluating the early integration of palliative care in patients with acute myeloid leukemia.

First Author	Study Design	Population	Intervention	Endpoints	Scales and Measures	Results
El-Jawahri A [65] 2020 and Nelson AM [68] 2021	Multisite, nonblinded, phase III randomized clinical trial	160 pts: 86 EPC 74 SC	EPC: **inpatient** PC physician, an AP nurse, or physician assistant. First visit within 72 of randomization. At least 2 visits a week during hospitalization up to 1 year after randomization. **No outpatient visits** SC: supportive care measures as per their oncology team. PC allowed at patients’ request or at the request of their oncologist.	**Primary**: QOL at week 2 **Secondary**: symptom burden, anxiety, depression, PTSD, patient reported EOL discussions, hospitalizations in the last week of life, chemotherapy in the last 30 days of life, and hospice use	FACT-Leuk ESAS PHQ-9 HADS PTSD Checklist–Civilia Brief COPE [68]	**Better QOL** (EPC:116.45 vs. SC:107.59; *p* = 0.04). **Lower depression** (EPC: 5.68 vs. SC: 7.20; *p* = 0.02; and EPC: 6.34 vs. SC: 8.00; *p* = 0.04). **Lower anxiety** (EPC: 4.53 vs. SC: 5.94; *p* = 0.02). **Lower PTSD symptoms** (EPC:27.79 vs. SC: 31.69; *p* = 0.01). **Greater use of approach-oriented coping** at 2 and 24 weeks (B = 1.85; SE = 0.62; *p* = 0.004 and B = –0.39; SE = 0.15; *p* = 0.01) [68]. **Lower use of avoidant coping** at week 2 (B = –0.70; SE = 0.29; *p* = 0.02) [68]. **Better QOL and lower anxiety, depression, and PTSD symptoms were maintained longitudinally**. **Higher frequency of discussion about EOL care preferences** (EPC: 21 of 28 [75.0%] vs. SC: 12 of 30 [40.0%]; *p* = 0.01) and **lower frequency of chemotherapy** in the last 30 days of life (EPC: 15 of 43 [34.9%] vs. SC: 27 of 41 [65.9%]; *p* = 0.01). **No differences** in symptom burden, PHQ-9 scores, or changes in the use of avoidant coping strategies [68], longitudinally. **No differences** in hospice use, hospice length of stay, or hospitalization in the last week of life.
Rodin G [66] 2020	Single-center phase II trial evaluating feasibility and tolerability, calculation of sample size, and timing of the primary endpoint	31 pts: 17 EPC 14 SC	EPC: **mainly inpatient** 8–12 psychotherapeutic sessions, over 8 weeks by a trained mental health clinician (EASE-psy), and systematic screening of physical symptoms (EASE-phys) with triggered referral to PC. PC team: a physician and nurse. First visit within 1 month of inpatient admission. **Rare outpatient evaluation** SC: PC allowed at request	**Primary**: severity of traumatic stress symptoms **Secondary**: physical symptom burden, pain, QOL, depressive symptoms and patients’ satisfaction with care	ESAS-AL SASRQ MSAS BPI FACIT- Sp BDI-II FAMCARE-P16	**Feasibility outcome met** **Reduced traumatic stress symptoms** at 4 and 12 weeks: EASE group: M (SE) = 24.26 (5.63), vs. SC group, M (SE) = 40.13 (5.50), *p* = 0.048; M (SE) = 21.03 (5.71), vs. SC group, M (SE) = 38.27 (5.46), *p* = 0.033 **Lower pain intensity and pain interference** with daily activities at 12 weeks, EASE group: M (SE) = 2.23 (2.66) vs. SC: M (SE) = 9.66 (2.09), *p* = 0.032. EASE group: M (SE) = 4.68 (6.27) vs. SC: M (SE) = 27.73 (4.88), *p* = 0.006. **Lower rates of pts with ASD** or threshold ASD at 12 weeks: EASE group: 7.7% (1/13) vs. SC: 42.1% (8/19), *p* = 0.05. **No differences** in physical symptom severity, symptom-related distress, depressive symptoms, satisfaction with care, and overall quality of life.
Potenza L [67] 2021	Single-center observational retrospective	215 pts: 131 EPC 84 late referrals to PC	EPC: **exclusively outpatient** One trained physician and one fellow First visit at a median of 5 weeks after the diagnosis. Monthly visits or frequency driven by disease trajectory. At least three visits Late PC: patients with only 1 or 2 visits of PC	**Primary**: presence of quality indicators of PC and EOL care	5 indicators of quality for PC [30]: psychological support, assessing and managing pain, GOC and prognosis, ACP, accessing home-care service 14 indicators of quality of EOL care [27]	**Higher rates of** Assessment and management of pain (EPC 100% vs. LatePC 46%; *p* = 0.00001) GOC (EPC 71.8% vs. LatePC 43%; *p* = 0.00001) ACP (EPC 57.3% vs. LatePC 2.3%; *p* = 0.00001) Home care service (EPC 43.5% vs. LatePC 14.2%; *p* = 0.00001) **Lower rate of** Chemotherapy in the last 14 days of life (EPC 2.7% vs. LatePC 13.9%; *p* = 0.0228) ICU admission and intubation in the last month of life (EPC 0% and 0% vs. LatePC 14.7% and 6.1%; *p* = 0.0007 and 0.0314) Access to ED ≥2 within 30 days of death (EPC 4% vs. LatePC 23.5%; *p* = 0.001) Death in acute facilities (EPC 5.3% vs. LatePC 31.4%; *p* = 0.002) RC transfusion in the last week of life (EPC 49.3% vs. LatePC 28.12%; *p* = 0.0315). **No differences** in Hospitalization ≥2 within 30 days of death, hospice length of stay > 7 days, platelet transfusion in the last week of life

EPC = early palliative care; SC = standard care; AP = advance practitioner; QOL = quality of life; PTSD = post-traumatic stress disorder; Brief COPE = Brief Coping Orientation to Problems Experienced Inventory; EOL = end-of-life; FACT-Leuk = Functional Assessment of Cancer Therapy–Leukemia; ESAS = Edmonton Symptom Assessment Scale; PHQ-9 = Patient Health Questionnaire; HADS = Hospital Anxiety and Depression Scale; EASE = Emotion And Symptom-focused Engagement; SASRQ = Stanford Acute Stress Reaction Questionnaire; MSAS = Memorial Symptom Assessment Scale; BPI = Brief Pain Inventory; FACIT-Sp = Functional Assessment of Chronic Illness Therapy-Spiritual Well-Being; BDI-II = Beck Depression Inventory-II; FAMCARE-P16 = Family Satisfaction with Care-Patient Version; GOC = goals of care conversations; ACP = advance care planning.

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
