# Peer review of "Early Palliative Care in Acute Myeloid Leukemia"

_cancers, 2022, doi:10.3390/cancers14030478_

Round 1
Reviewer 1 Report
The review is well written. This is an important topic and the information described can raise awareness to clinicians for early integration of PC in AML care.
The challenges of introducing PC in hematological malignancies are clearly delineated.
Reviewer 2 Report
Lines 73-74 and lines 76-86: please detail – eg.: relevant studies regarding the outcomes of these emerging therapies in terms of disease trajectory and quality of life / quality of care, eventually focused on AML patients
Line 211: Please respect the abbreviation: AML
Lines 520-521: I would rather use the term “palliative care specialist physician”, not “early palliative care physician”. I would also suggest the introduction of a palliative care module in the training curriculum for oncology or hematology specialties, as well as for other medical specialties.
The authors conducted an interesting and well-documented review of the literature highlighting the benefits of EPC in haematological malignancies, suggesting that the integrated model of EPC could be extended to AML patients.
I also appreciate the originality in the elaboration of the figure "Effects of the early integration of palliative care on the disease trajectory of patients with Acute Myeloid Leukemia", as well as the highlighting of future challenges.
In fact, from my experience as a palliative care physician, I am convinced of the potential benefits of promoting and implementing an integrated model of early palliative care in oncological care for patients with both solid tumors and haematological malignancies.
